# Growth and Leaf Gas Exchange Upregulation by Elevated [CO_2_] Is Light Dependent in Coffee Plants

**DOI:** 10.3390/plants12071479

**Published:** 2023-03-28

**Authors:** Antonio H. de Souza, Ueliton S. de Oliveira, Leonardo A. Oliveira, Pablo H. N. de Carvalho, Moab T. de Andrade, Talitha S. Pereira, Carlos C. Gomes Junior, Amanda A. Cardoso, José D. C. Ramalho, Samuel C. V. Martins, Fábio M. DaMatta

**Affiliations:** 1Departamento de Biologia Vegetal, Universidade Federal de Viçosa, Viçosa 36570-900, MG, Brazil; 2Department of Crop and Soil Sciences, North Carolina State University, Raleigh, NC 27695, USA; 3PlantStress & Biodiversity Lab., Centro de Estudos Florestais (CEF), Laboratório Associado Terra, Departamento de Recursos Naturais, Ambiente e Território (DRAT), Instituto Superior de Agronomia (ISA), Universidade de Lisboa (ULisboa), Quinta do Marquês, Av. da República, 2784-505 Oeiras, Portugal; 4Unidade de Geobiociências, Geoengenharias e Geotecnologias (GeoBioTec), Faculdade de Ciências e Tecnologia (FCT), Universidade NOVA de Lisboa (UNL), Monte de Caparica, 2829-516 Caparica, Portugal

**Keywords:** climate change, elevated [CO_2_], nitrogen, photosynthetic down-regulation, shading, stomatal conductance

## Abstract

Coffee (*Coffea arabica* L.) plants have been assorted as highly suitable to growth at elevated [CO_2_] (e*C*_a_), although such suitability is hypothesized to decrease under severe shade. We herein examined how the combination of e*C*_a_ and contrasting irradiance affects growth and photosynthetic performance. Coffee plants were grown in open-top chambers under relatively high light (HL) or low light (LL) (9 or 1 mol photons m^−2^ day^−1^, respectively), and a*C*_a_ or e*C*_a_ (437 or 705 μmol mol^–1^, respectively). Most traits were affected by light and CO_2_, and by their interaction. Relative to a*C*_a_, our main findings were (i) a greater stomatal conductance (*g*_s_) (only at HL) with decreased diffusive limitations to photosynthesis, (ii) greater *g*_s_ during HL-to-LL transitions, whereas *g*_s_ was unresponsive to the LL-to-HL transitions irrespective of [CO_2_], (iii) greater leaf nitrogen pools (only at HL) and higher photosynthetic nitrogen-use efficiency irrespective of light, (iv) lack of photosynthetic acclimation, and (v) greater biomass partitioning to roots and earlier branching. In summary, e*C*_a_ improved plant growth and photosynthetic performance. Our novel and timely findings suggest that coffee plants are highly suited for a changing climate characterized by a progressive elevation of [CO_2_], especially if the light is nonlimiting.

## 1. Introduction

The atmospheric CO_2_ concentration (*C*_a_) has progressively increased since the industrial revolution and currently approaches 420 μmol mol^−1^. Plants under elevated *C*_a_ (e*C*_a_) often display decreased stomatal density (SD) and aperture, potentially reducing stomatal conductance (*g*_s_) and leaf transpiration [1,2]. Despite the lower *g*_s_, the net rate of CO_2_ assimilation (*A*) increases at e*C*_a_ due to a greater CO_2_ supply to the catalytic sites of ribulose-1,5-bisphosphate carboxylase/oxygenase (RuBisCO), which uplifts RuBisCO carboxylation over oxygenation [1,2]. Concurrently, rates of photorespiration and transpiration per unit leaf area decrease, whereas water- and light-use efficiency, as well as biomass accumulation are improved [2,3,4]. However, greater biomass under e*C*_a_ can bring about nutrients and minerals “dilution” in plant tissues, especially nitrogen (N) [5,6,7]. In addition, impairments in N uptake and assimilation suppression at e*C*_a_ may also exacerbate the decrease of shoot N pools [8]. In the long-term, lower N levels may decrease the investment in RuBisCO coupled to a lower maximum apparent carboxylation capacity (*V*_cmax_) and these decreases are often associated with a build-up of nonstructural carbohydrate (NSC) pools [2]. Collectively, these alterations may lead to a decreased photosynthetic capacity known as photosynthetic down-regulation [9].

Coffee (*Coffea arabica* L.) is one of the most important traded commodities on a global scale. This species evolved in the understories of African highland forests, but it has been cultivated worldwide over distinct light environments, ranging from full sun exposure (e.g., Brazil) to relatively dense shade (e.g., some regions of Central America and India). Coffee plantations at full sunlight often outyield those under shade in regions with satisfactory edaphoclimatic conditions for crop production [10,11]. Accordingly, both *A* and *g*_s_, as well as mesophyll conductance (*g*_m_), are greater in high light (HL) than in low light (LL) conditions, particularly when vapor pressure deficit (VPD) is low [10,11,12]. However, given the coffee’s origin in understory forests, unshaded plants are believed to be under a higher threat in face of environmental stresses (e.g., drought and extreme temperatures) than their shaded counterparts. If that holds true, the negative consequences of climate change and global warming on coffee plantations will be considerably magnified in full sunlight coffee farms [13]. Under this perspective, the use of shelter trees in coffee plantations, namely with the implementation of tree intercropping or agroforestry systems, has been routinely invoked as a management strategy to ameliorate the sustainability of the coffee crop in a climate change scenario [11,14,15]. Notably, a recent meta-analysis (without considering the effects of e*C*_a_), has suggested that coffee yields tended to be the highest in full sunlight or in “low to moderate” shaded environments (no more than 40% of shade cover), while “very high” shade (above 70%) led to the lowest yields for most coffee cultivars [14].

Irrespective of the light levels, coffee plants have been assorted as highly susceptible to several stresses at current ambient *C*_a_ (a*C*_a_) [16,17,18]. Nevertheless, a greater intrinsic resilience of some coffee genotypes than anticipated, as well as the positive impact of e*C*_a_, can significantly attenuate stress impairments, namely of supraoptimal temperatures [19,20,21], and drought [22,23,24,25,26,27] in coffee. This mitigation by e*C*_a_ has been linked to an amplified acclimation response associated with improved antioxidant system and other protective molecules, together with greater lipid dynamics, and enhanced photosynthetic performance with no apparent photosynthetic down-regulation [20,21,22,28,29,30,31]. Interestingly, in contrast to most species so far investigated, which often display marked reductions in *g*_s_ at e*C*_a_ (see [2,32]), *g*_s_ in coffee seems to be mostly unresponsive to e*C*_a_ [20,22,28,30]. In fact, in some cases, *g*_s_ has even been shown to increase at e*C*_a_, as circumstantially noted in free-air CO_2_ enrichment (FACE) trials in the upper (but not in the lower) canopy leaves that are more exposed to direct sunlight [31]. More recently, Marçal et al. [33] compared coffee plants that were grown under two light conditions [moderate LL (53% light restriction) and HL (no light restriction)] and two *C*_a_ conditions [ambient *C*_a_ (a*C*_a_; 400 µmol CO_2_ mol^−1^) and e*C*_a_ (700 µmol CO_2_ mol^−1^); the authors found that *g*_s_ did not respond to the *C*_a_ *per se*, but *g*_s_ was only significantly higher in HL at e*C*_a_. This lack of response of *g*_s_ to light at a*C*_a_ contrasts with several studies showing higher *g*_s_ (and *A*) in HL over LL when shading is more severe (e.g., 90% light restriction) [12,34,35]. At a first glance, these findings suggest an interplay between light intensity and *C*_a_ on stomatal function in coffee, with consequences to the photosynthetic performance and ultimately plant growth.

The understanding of the underlying mechanisms by which coffee plants cope with light and *C*_a_ supplies at both the leaf and whole-plant scales is of paramount importance for crop sustainability in face of ongoing climate changes. In a first attempt to understand these mechanisms, Marçal et al. [33] have demonstrated that young coffee plants grown under moderate LL displayed lower growth and photosynthetic performance relative to those grown under HL. These impacts were partially reversed by an e*C*_a_, although most effects of light and *C*_a_ on growth and photosynthetic performance occurred independently [33]. Here, we further explored the mechanisms by which the photosynthetic machinery adjusts to more contrasting light conditions (with a reduction of irradiance up to 90%, LL), and how these adjustments could be modulated under e*C*_a_. Specifically, our main objectives were to examine how the interplay of light levels and *C*_a_ could affect (i) the stomatal function, (ii) the relative contributions of photochemical, diffusional, and biochemical limitations of photosynthesis at the leaf level, (iii) the N partitioning into the major components of the photosynthetic apparatus, (iv) as well as the coffee growth performance.

## 2. Results

Most variables analyzed remarkably responded to light and/or C_a_. Key traits of leaf gas exchange and photosynthesis [*A*, *g*_s,_ and electron transport rate (ETR)] and growth [total biomass (TB) and total leaf area (TLA)] were not only affected by light and *C*_a_ factors, but also by their interaction (*p* < 0.001) (Table A1).

### 2.1. Gas Exchanges and Chlorophyll a Fluorescence Parameters

Shading markedly affected the magnitude of gas exchange, with lower *A* in LL plants (e*C*_a_: 49%, a*C*_a_: 51%) than those of HL ones (values within parentheses throughout the results represent the percentage changes of a given treatment with regards to its direct counterpart). The lower *A* was accompanied by lower values of *g*_s_ (e*C*_a_: 66%, a*C*_a_: 53%), transpiration rate (*E*) (e*C*_a_: 76%, a*C*_a_: 68%), nocturnal respiration rate (*R*_n_) (e*C*_a_: 45%, a*C*_a_: 28%), and photorespiration-to-gross photosynthesis ratio (*R*_P_/*A*_G_) (e*C*_a_: 34%, a*C*_a_: 16%) relative to HL plants (Figure 1).

Regardless of irradiance, *A* more than doubled in e*C*_a_ relative to a*C*_a_ plants (HL: 112%, LL: 124%), paralleling significant increases in internal *C*_a_ (*C*_i_) (HL: 92%, LL: 72%). As expected, e*C*_a_ decreased *R*_P_/*A*_G_ (HL: 47%, LL: 59%). Notably, *g*_s_ (as also *R*_n_) from HL/e*C*_a_ plants was ca. 25% higher than that of their HL/a*C*_a_ counterparts, but no differences were found between *C*_a_ conditions under LL (Figure 1).

The LL plants, when evaluated at 1000 µmol photons m^−2^ s^−1^ (Figure 2), displayed increases in *A* (e*C*_a_: 45%, a*C*_a_: 44%), *g*_s_ (e*C*_a_: 60%, a*C*_a_: 51%), ETR (e*C*_a_: 59%, a*C*_a_: 60%) and ETR/*A* (e*C*_a_: 23%, a*C*_a_: 39%), as compared with their values obtained at growth irradiance (Figure 1). Conversely, the HL plants when evaluated at 200 µmol photons m^−2^ s^−1^ showed, in general, greater reductions in those parameters (especially in *g*_s_) at a*C*_a_ than at e*C*_a_: decreases in *A* (e*C*_a_: 17%, a*C*_a_: 36%), *g*_s_ (e*C*_a_: 15%, a*C*_a_: 67%), ETR (e*C*_a_: 45%, a*C*_a_: 51%), and ETR/*A* (e*C*_a_: 10%, a*C*_a_: 19%) (Figure 2).

The maximum photochemical efficiency of photosystem II (*F*_v_*/F*_m_) ratio was unresponsive to the treatments and averaged close to 0.8. LL plants, as compared to HL ones, exhibited greater values of the actual photochemical efficiency of photosystem II (*F*_v_′/*F*_m_′) (e*C*_a_: 6%, a*C*_a_: 8%), the actual quantum yield of PSII (Φ_PSII_) (e*C*_a_: 129%, a*C*_a_: 152%), and *q*_p_ (e*C*_a_: 84%, a*C*_a_: 115%), but with substantially lower ETR values (e*C*_a_: 59%, a*C*_a_: 51%). That resulted in a lower ETR/*A* (e*C*_a_: 21%, a*C*_a_: 8%) (Table A2).

In HL individuals, e*C*_a_ promoted an increase in Φ_PSII_ (13%) and ETR (23%), whereas in LL plants these parameters were unresponsive to e*C*_a_. In contrast, e*C*_a_ led to reductions in the ETR/*A* ratio (HL: 44%, LL: 52%) (Table A2).

### 2.2. A/C_i_ Curves

The LL plants showed lower values of *V*_cmax_ (e*C*_a_: 15%, a*C*_a_: 16%), maximum rate of carboxylation that is limited by electron transport (*J*_max_) (e*C*_a_: 15%, a*C*_a_: 17%), and *g*_m_ (e*C*_a_: 32%, a*C*_a_: 37%) relative to the HL ones. Regardless of treatments, photosynthetic performance was mainly constrained by diffusive components [i.e., stomatal limitation (*l*_s_) + mesophyll limitation (*l_m_*); ≈67%]. Notably, the LL plants, regardless of *C*_a_, showed higher *l_s_* (13% on average) than those of HL individuals (Figure 3).

The e*C*_a_ promoted overall increases in *V*_cmax_ (HL: 27%, LL: 28%) and *J*_max_ (HL: 25%, LL: 28%). The *J*_max_/*V*_cmax_ ratio did not respond to the treatments and averaged on ca. 2.0, as can be deduced from Figure 3.

### 2.3. Photosynthetic N-Use Efficiency and N Partitioning

The HL/e*C*_a_ plants had the highest leaf N concentration on both a per mass and area bases (37.9 g kg^−1^ and 2.16 g m^−2^, respectively) relative to plants from the other treatments (averaging on 32 g kg^−1^ and 1.64 g m^−2^; Table 1). Overall, the LL plants invested approx. half of their total leaf N pools in photosynthesis (*P*_P_) against ca. 41% in HL plants, irrespective of *C*_a_ (Table 1). Accordingly, LL plants, relative to their HL counterparts, displayed higher leaf N fraction allocated to RuBisCO (*P*_R_) (e*C*_a_: 23%, a*C*_a_: 17%), leaf N fraction allocated to bioenergetics (*P*_B_) (e*C*_a_: 20%, a*C*_a_: 42%) and leaf N fraction allocated to thylakoid light-harvesting components (*P*_L_) (e*C*_a_: 27%, a*C*_a_: 12%, significant only for e*C*_a_), implying in concordant decreases in leaf N fraction allocated to non-photosynthetic tissues (*P*_NP_) (e*C*_a_: 17%, a*C*_a_: 11%). Photosynthetic-N use efficiency (PNUE) did not respond to light availability in e*C*_a_ plants but increased by 34% in a*C*_a_/HL individuals relative to a*C*_a_/LL ones (Table 1).

The e*C*_a_ condition promoted an increase in *P*_P_ (10%) in LL plants with a concomitant increase only in *P*_R_ (14%). In HL plants e*C*_a_ did not affect *P*_P_, despite the increments in *P*_B_ (23%), whereas *P*_L_ and *P*_R_ were unresponsive to C_a_. Regardless of light, PNUE strongly responded to e*C*_a_, with increases of 68% (HL) and 133% (LL) (Table 1).

### 2.4. Metabolites

The LL plants, compared with HL ones, had higher concentrations of total chlorophylls (Chl) (e*C*_a_: 52%, a*C*_a_: 23%), as well as lower concentrations of starch (e*C*_a_: 11%, a*C*_a_: 17%), glucose (e*C*_a_: 31%, a*C*_a_: 14%), fructose (e*C*_a_: 27%, a*C*_a_: 12%), NSC (e*C*_a_: 12%, a*C*_a_: 14%) and total amino acids (TAA) (e*C*_a_: 31%, a*C*_a_: 40%) (Table 2).

In LL plants, e*C*_a_ led to a decline in fructose (11%) concentration, and increases in sucrose (17%), TAA (79%), and proteins (20%) pools. In HL plants, e*C*_a_ caused increases in the levels of TAA (61%), proteins (21%), glucose (20%), and fructose (7%) (Table 2).

### 2.5. Morphoanatomical Traits

The LL individuals displayed lower values of SD (e*C*_a_: 53%, a*C*_a_: 51%), stomatal index (SI) (e*C*_a_: 26%, a*C*_a_: 17%), and maximum theoretical stomatal conductance to water vapor (*g*_wmax_) (e*C*_a_: 56%, a*C*_a_: 49%) than those of HL ones. These traits were unaffected by *C*_a_, except *g*_wmax_ which was 21% higher in e*C*_a_ than in a*C*_a_ plants at HL. The stomatal size (SS) did not respond significantly to the treatments, although a tendency to increase (20%) was observed in e*C*_a_ when compared with a*C*_a_, only under HL conditions (Table 3).

### 2.6. Growth Traits and Biomass Partitioning

Total biomass (TB) and total leaf area (TLA) were markedly affected by light and *C*_a_ (*p* < 0.001), and by their interaction (Table A1). Light availability did not significantly affect root mass ratio (RMR), stem mass ratio (SMR) and leaf mass ratio (LMR), in contrast to the *C*_a_ factor, which increased RMR at the expense of reductions in LMR (Table 4). Despite unchanging SMR, qualitative changes were observed when this SMR was broken down into orthotropic mass ratio (OMR) (which varied negligibly in response to the treatments) and plagiotropic mass ratio (PMR): PMR was higher in HL than in LL conditions and increased in response to e*C*_a_ (Table 4).

The LL conditions provoked dramatic declines in growth and TB, especially at a*C*_a_ (Table 4). Relative to HL plants, the LL ones showed reductions in height (e*C*_a_ and a*C*_a_: 47%) and TLA (e*C*_a_: 71%, a*C*_a_: 68%), with concordant lower TB (e*C*_a_: 84%, a*C*_a_: 79%), although with greater leaf area ratio (LAR) (e*C*_a_: 80%, a*C*_a_: 63%) (Table 4).

Plant phenotypes were significantly altered by e*C*_a_, as deduced from the greater height (HL and LL: 23%), TLA (HL: 62%, LL: 42%), and TB (HL: 102%, LL: 53%) than at a*C*_a_ (Table 4). RMR increased dramatically (HL: 103%, LL: 41%) at the expense of a lower LMR (HL: 21%, LL: 10%), whereas SMR and LAR remained unchanging independently of *C*_a_ (Table 4).

## 3. Discussion

We here demonstrated that the positive effects of eC_a_ on coffee performance were particularly evident under moderate to high light levels. In any case, we must contend that the “HL” conditions herein employed were approx. half relative to that found outdoors in the same experimental site [34], and thus our HL treatment is more moderate relative to agricultural conditions. Notably, however, we must emphasize that (i) the usual levels of shade cover for coffee cultivation often range from 20 to 50% [10,13], and that (ii) young plants remarkably display less self-shading than adult individuals, implying that the levels of proper shade cover could exceed those for adult coffee plants. We, therefore, contend that, within the given limits, our major findings may be properly applied to agricultural conditions such as agroforestry systems.

### 3.1. The eC_a_ Increases Stomatal Conductance Only in High-Light-Grown Plants

Despite having evolved in shady environments, coffee plants often display higher *g*_s_ (associated with higher SD and SI) at HL than at LL conditions [10,12,33], in agreement with our results. The dependence of stomatal aperture on light intensity appears to be independent of *C*_a_, as noted both under moderate irradiance [28,34] and a severe shade (this study). Notably, we determined *g*_s_ using an infrared gas analyzer and the *g*_wmax_ using anatomical traits [36] and both traits responded similarly to the applied treatments. However, as there were no significant variations in SD, SI, and SS (as well as in unitary leaf area; data not shown), in good agreement with our previous reports [20], we contend that the stomatal aperture played major roles in governing the changes in *g*_s_ in response to light and C_a_. Most importantly, we herein showed for the first time an increase in *g*_s_ at e*C*_a_ in HL plants (Figure 1). Similar results were also observed in field-grown coffee trees by Rakočević et al. [31], as well as in some woody taxa adapted to warm, low-humidity conditions [37]. Taking all these findings together, we suggest that responses of *g*_s_ to e*C*_a_ might be species-specific, implying that some caution should be exercised against the well-established universal belief that *g*_s_ is downregulated by e*C*_a_ [1,2,3].

The interactive effects of light and CO_2_ availability on stomatal behavior were immediately evident when we assessed gas exchanges of HL plants under LL (and vice-versa) (Figure 2). During the HL-to-LL transitions, we observed decreases in *g*_s_, but to a much lesser extent in e*C*_a_ plants than in a*C*_a_ individuals; in contrast, in the LL-to-HL transitions we found increases in *g*_s_ that seemed independent of C_a_. Although the underlying physiological mechanisms that could explain this differential stomatal behavior remain to be elucidated, our results have direct practical implications and help explain the greater carbon gain (and biomass) over time in the HL/e*C*_a_ individuals. It is important to emphasize that coffee plants display inherently low *g*_s_ and stomatal constraints comprise the main limitation to photosynthesis, particularly in LL leaves (Figure 1 and Figure 3) [12]. In this context, maintenance of higher *g*_s_ under partial shading can guarantee a greater CO_2_ inflow, ultimately improving whole-canopy photosynthesis [38]. In fact, *A* was a little constrained during the HL-to-LL transition in e*C*_a_ plants (Figure 2). Extrapolating this information to the field, where the intercepted irradiance along the day varies dramatically throughout the coffee canopy [39], an e*C*_a_ would favor carbon gain via higher *g*_s_ when the foliage becomes momentarily more shaded. This might partly explain the improved coffee growth and production at e*C*_a_, as observed in FACE-grown coffee trees [11,29]. Additionally, higher *g*_s_ at e*C*_a_ may result in lower leaf temperature (data not shown), which may translate into lower rates of maintenance respiration and photorespiration [2], further contributing to improving carbon balance and biomass accumulation.

### 3.2. The eC_a_ Improves Photosynthetic Performance by Decreasing Diffusional Limitations without Causing Photosynthetic Down-Regulation

As already demonstrated in previous studies with coffee [12,30,34], diffusive limitations (*l*_s_ + *l*_m_) comprised the largest fraction of total constraints to photosynthesis, as can be deduced from Figure 3. LL plants showed lower *g*_s_ in parallel to higher *l*_s_ than HL plants, regardless of C_a_. In LL plants, since there were no significant differences in *g*_s_ and *g*_m_ (and *l*_m_) in response to *C*_a_, the CO_2_ enrichment resulted in higher *A* likely associated with a higher CO_2_ availability in the chloroplasts and hence lower photorespiration. However, it should be noted that the *l*_s_ of LL plants under e*C*_a_ was even greater than the *l*_s_ of their HL counterparts (Figure 3), underlining the limitations imposed by the lower *g*_s_ on the photosynthetic performance by LL conditions. Indeed, another indicator of the magnitude of the diffusive limitations in LL plants, in both CO_2_ conditions, was the highest values of ETR/*A* ratio when these plants were evaluated under HL (Figure 2), indicating a higher presence of alternative electron sinks.

Overall, both HL and particularly e*C*_a_ provoked adjustments in RuBisCO activity (as noted by *V*_cmax_) and RuBP regeneration capacity (*J*_max_) (Figure 3). However, *V*_cmax_ and *J*_max_ varied in parallel regardless of treatments, resulting in an unchanged *J*_max_/*V*_cmax_ ratio, in contrast to meta-analysis studies which have shown that such a ratio increased slightly (5.7% on average) but significantly at e*C*_a_ [9,40]. In addition, in good agreement with our previous studies [22,28,30,34], we found no signs of photosynthetic acclimation at e*C*_a_, as denoted by the up-, rather than down-regulation, of *V*_cmax_ and *J*_max_, lack of decreases in N levels and relatively small changes in NSC pools. In fact, increased *A* values in e*C*_a_ coffee plants have been shown to be supported not only by the greater CO_2_ availability at the carboxylation sites (with concordant decreases in photorespiration), but also by greater in vivo thylakoid electron transport rates at both photosystems I and II levels, and some further investment in photosynthetic components (among them RuBisCO) [20,26]. Most importantly, photosynthetic up-regulation at e*C*_a_ was herein observed even when growth was severely impaired (low sink demand) by LL conditions (see below), which additionally lends support for the suitability of coffee plants in a scenario of increasing *C*_a_.

### 3.3. The eC_a_ Improves Photosynthetic-N Use Efficiency Irrespective of Light Levels via Increases in Photosynthetic Performance Rather Than Alterations in N Partitioning

Most studies have reported unchanging foliar N pools in coffee at e*C*_a_ [29,30,34], although a “dilution effect” of N was also observed in a genotype-dependent manner [10]. Unexpectedly, we noted an increased N concentration in HL/e*C*_a_ plants (Table 1), which might be associated with improved N acquisition due to increased soil exploitation (higher RMR, Table 4), and higher N mass flow to the foliage due to the greater *E* (Figure 1) in those plants. We also noted a higher protein concentration under e*C*_a_ (and even greater as regards amino acids) irrespective of the light regime, suggesting a differential partitioning of N towards protein synthesis. Since the greatest proportion of leaf proteins is associated with RuBisCO [41], we suggest that e*C*_a_ improved RuBisCO content, which is consistent with the e*C*_a_-related increases in *V*_cmax_ in both LL plants (unchanging foliar N but with increased *P*_R_) and HL ones (increased N but with unchanging *P*_R_), in line with the above-mentioned reports that found a significant up-regulation of total RuBisCO activity in e*C*_a_ coffee plants grown at moderate irradiance [20,26].

The marked e*C*_a_-related improvement in PNUE found here (68–133%) was much greater than the average increase (31%) reported by Leakey et al. [4] when analyzing several species grown under FACE conditions. Despite being improved markedly by e*C*_a_, PNUE in coffee is quite low in comparison with other woody species [42,43,44]. At a first glance, such a low PNUE in coffee leaves might be linked to a poor investment of N (41–52%) in the photosynthetic machinery [45,46] and, especially, due to the inherently low *A*. Given that the greater *P*_P_ in LL plants was not translated into improved PNUE relative to that of the HL individuals, we suggest that varying patterns of N allocation to photosynthesis had a minor role in explaining the intrinsic low coffee’s PNUE. Furthermore, to maximize PNUE at e*C*_a_, it has been proposed that leaf N pools should be reallocated from RuBisCO (thus leading to a lower *P*_R_) to the RuBP regeneration (higher *P*_B_) and/or light-harvesting pigments (higher *P*_L_) [47]. Here we observed that e*C*_a_ provoked increases in *P*_R_ without affecting *P*_B_ or *P*_L_ only in LL plants, whereas in HL individuals e*C*_a_ led to increases in *P*_B_ with unchanging *P*_R_ (Table 1). Altogether, we contend that e*C*_a_ could improve PNUE by fundamentally decreasing the diffusive limitations of photosynthesis (and decreasing *R*_p_) rather than by altering N allocation to the photosynthetic apparatus.

### 3.4. The eC_a_ Interacts with High Light to Promote Growth, but Changes in Biomass Partitioning Are Only Affected by the CO_2_ Supply

Both higher light and CO_2_ availabilities improved biomass gain (and *A*), and the positive effects of e*C*_a_ were much less evident in LL conditions. These results suggest that the revenues in terms of growth and photosynthetic performance at e*C*_a_ are expected to be decreased at deep shade, especially in coffee plants with dense canopies grown at closer spacings. We also found that the overall biomass partitioning between roots, leaves, and stems was unaffected by light, which accords with our previous findings [34]. However, in sharp contrast with results reported by Marçal et al. [33], who observed an unchanged partitioning in response to varying *C*_a_, we herein found an increased biomass partitioning to roots (higher RMR) at the expense of leaves (lower LMR) at e*C*_a_ (Table 4). Possibly, differences in the age of plants (six-month-old plants in [33] against 15-month-old in this current study) could result in ontogenetic drifts that might explain these apparent discrepancies (see [34,48]). In any case, increases in RMR at e*C*_a_ have been documented in other woody species such as eucalyptus [49] and could significantly improve water and nutrient acquisition [2,48], thus leading to better fitness, as particularly noted in HL/e*C*_a_ plants. Interestingly, e*C*_a_ provoked earlier branching, especially in HL, thus allowing a greater node formation that in turn could anchor a greater number of leaves. Given that the unitary leaf area was unaffected by e*C*_a_ (in contrast to the general increase that has been observed at e*C*_a_) (see [2]), we contend that the greater total leaf area was to a large extent associated with more nodes per branch. Additionally, if the number of nodes is the key component of coffee production [10], crop yields should then increase at e*C*_a_, as has been reported for FACE-grown coffee trees [11,29].

Plant biomass can be regarded as the most direct measure of performance as a product of growth [50]. Biomass accumulation increases with increasing relative growth rate, which is the product of LAR and net assimilation rate (NAR) (which in turn is directly associated with the time-integrated whole-plant *A*) [51]. The increases in biomass we found were accompanied by decreased LAR (HL conditions) or unchanging LAR (e*C*_a_ conditions). Thus, it is tempting to suggest that physiological adjustments, leading to a higher NAR, should greatly explain the differences in biomass observed due to the varying supplies of both light and CO_2_. If computing solely the effects of e*C*_a_ on growth under non-limiting light, we found in e*C*_a_ relative to a*C*_a_ individuals: (i) a remarkably greater biomass (107%) with the same LAR; (ii) a higher proportion of non-photosynthetic tissues (48% against 35%; as deduced from the biomass partitioning among leaves, stems and roots), which consume but do not produce photoassimilates, thus contributing negatively to the whole-plant carbon balance at an e*C*_a_ condition; and (iii) no differences in foliar NSC pools. Collectively, this information reinforces the prominent role of photosynthesis, and probably NAR, to support the higher growth rates and biomass gain in an e*C*_a_ scenario when light is not limiting. At last, we speculate to what extent the higher RMR under e*C*_a_ improved water acquisition thus allowing a longer stomatal aperture, and, consequently, increased the time-integrated whole-plant *A*. In this context, whereas the higher RMR seems advantageous for HL/e*C*_a_ plants, it might be a penalty for LL/e*C*_a_ plants that would benefit more from an investment in TLA prioritizing light capture.

## 4. Materials and Methods

### 4.1. Plant Material, Growth Conditions, and Sampling

The experiment was carried out in Viçosa (20°45′17″ S, 42°52′57″ W, 650 m above sea level), southeastern Brazil. Coffee seedlings (*C. arabica* cv. Catuaí Vermelho IAC 44, obtained from the “Empresa de Pesquisa Agropecuária de Minas Gerais”—EPAMIG, Viçosa, Minas Gerais State, Brazil) with four leaf pairs were transplanted into 12-L pots containing soil, sand, and composted manure (3:2:1; *v*:*v*:*v*). This substrate, with a pH of 5.8, had the following macronutrient composition: N, P, K, S (600, 12.4, 179, 53.7 mg kg^−1^, respectively), and Ca and Mg (0.97 and 0.62 cmol_c_ dm^−3^, respectively). The substrate was fortnightly fertilized with 2 g of ammonium sulfate and 100 mL of Hoagland’s nutrient solution [52]. When 3-month-old (5–6 leaf pairs), 28 plants (one plant per pot) were distributed among four open top chambers (OTC) (1.15 m in diameter and 1.40 m in height). The *C*_a_ levels were 437 ± 9 μmol mol^–1^ (a*C*_a_) or 705 ± 18 μmol mol^–1^ (e*C*_a_). Detailed information on CO_2_ enrichment (from 06:00 to 18:00 h) and *C*_a_ checking inside the OTCs have been reported by Ávila et al. [22]. The OTCs were maintained in a greenhouse with controlled temperature (ca. 30/25 °C, day/night) and naturally fluctuating conditions of air humidity and irradiance. After 150 days in the OTCs, the least uniform plant inside each OTC was discarded and the remaining six plants in individual pots were maintained in each chamber. During the entire period over which the plants remained inside the OTCs, plants were subjected to two irradiance levels (in either aCO_2_ and eCO_2_), as measured inside the OTCs: (i) HL [natural, direct photosynthetic photon flux density (PPFD) passed on by the greenhouse (approx. 9.0 mol photons m^−2^ day^−1^)], and (ii) LL [ca. 90% restriction of HL level, which gave approx. 1.0 mol photons m^−2^ day^−1^ at the top of the plants]. The light restriction was obtained using neutral-density black nylon nettings (Sombrite^®^). PPFD, air temperature (T_air_), and air relative humidity inside the OTCs were registered with sensors (positioned above the plant canopies) every minute and stored as 15 min averages in a data logger (Li-1400, Li-Cor, Lincoln, NE, USA). Over the course of the experiment, the diurnal T_air_ averaged 24.3 and 22.8 °C, whereas VPD averaged 1.03 and 0.84 kPa, respectively in HL and LL conditions. Both T_air_ and VPD were unaffected by *C*_a_ within each light environment. The pots inside the OTCs were rotated weekly to minimize any spatial variation in light and CO_2_.

Sampling and measurements were carried out in October and November 2021, when plants were approximately 15 months old. For anatomical and physiological analyses, fully expanded leaves from the third or fourth node from the apex of the plagiotropic branches were used.

### 4.2. Gas Exchanges and Chlorophyll a Fluorescence Analysis

Leaf gas exchanges and Chl *a* fluorescence parameters were concurrently assessed using a gas exchange system equipped with an integrated fluorescence chamber (LI-6400XT, LI-Cor, Lincoln, NE, USA). The *A*, *g*_s_, *C*_i_, and *E* were measured under artificial PPFD of 1000 µmol m^−2^ s^−1^ for HL plants, and 200 µmol m^−2^ s^−1^ for LL plants. These PPFD intensities resembled roughly the maximum ambient PPFD intercepted by the sampled leaves in their natural angles during evaluations (the LL plants had more planophile leaves than the HL ones, thus intercepting the incident light beams more efficiently). Measurements were also conducted under a PPFD of 200 µmol m^−2^ s^−1^ for HL and a PPFD of 1000 µmol m^−2^ s^−1^ for LL plants. The amount of blue light was fixed at 10% of PPFD to favor the stomatal aperture. The block temperature was maintained at 25 °C, and the VPD was kept at ca. 1.0 kPa. All the evaluations, including *R*_n_ (see below), were conducted at either 400 or 700 µmol CO_2_ mol^−1^, according to each CO_2_ treatment. Corrections for the leakage of CO_2_ into and out of the leaf chamber of the LI-6400XT were carried out for all gas exchange gauges [53]. Additional details are described in DaMatta et al. [30].

The Chl *a* fluorescence parameters were measured in both dark- and light-acclimated leaf tissues exactly as described elsewhere [34], from which *F*_v_/*F*_m_, *F*_v_′/*F*_m_′, Φ_PSII_, and ETR were calculated.

The *R*_n_ was measured at midnight using the same gas exchange system described above and divided by two (*R*_n_/2) to estimate the daytime mitochondrial respiration rate (*R*_l_) [54]. The *R*_l_ values were rectified for leaf temperature using the temperature response equations given by Sharkey et al. [55]. The RuBisCO photorespiratory rate (*R*_P_) was calculated as *R*_P_ = 1/12[ETR − 4(*A* + *R*_l_)] following [56], after which *R*_P_/*A*_G_ was computed by considering the values of *R*_P_, *A,* and *R*_l_ [30]. Afterward, *g*_m_ was estimated according to the methodology outlined by [57], for which gas exchange (*A*, *C*_i,_ and *R*_l_) and Chl *a* fluorescence (ETR) data were used. Additionally, the conservative parameter Γ* (chloroplastic CO_2_ compensation point in the absence of mitochondrial respiration) for coffee was taken from Martins et al. [58]. The *g*_m_ was then calculated as follows:*g*_m_ = *A*/(*C*_i_ − (Γ*(ETR + 8(*A* + *R*_l_))/(ETR − 4(*A* + *R*_l_))))

Six in situ *A*/*C*_i_ curves, produced in the morning exactly as described elsewhere [12,30,34], were performed under a PPFD of 1000 μmol m^–2^ s^–1^ at the leaf level, which is sufficient to saturate the photosynthetic machinery of coffee leaves without causing photoinhibition [40]. From these curves, *V*_cmax_ and the maximum rate of carboxylation limited by electron transport (*J*_max_), both on a *C*_c_ basis, were calculated by fitting the mechanistic model of CO_2_ assimilation described by Farquhar et al. [59] using the *C*_c_-based temperature dependence of the kinetic parameters of RuBisCO [60]. The values of *V*_cmax_, *J*_max,_ and *g*_m_ were normalized to 25 °C using the temperature response equations proposed by Sharkey et al. [55].

The overall photosynthetic limitations, fractionated into their functional components [stomatal (*l*_s_), mesophyll (*l*_m_), and biochemical (*l*_b_)], were estimated following the equations given in Grassi and Magnani [61] and comprehensively described by Martins et al. [12].

### 4.3. Nitrogen-Use Efficiency and N Partitioning

Total leaf N was determined using the Kjeldahl method as reported elsewhere [62]. The PNUE was estimated from the ratio of light-saturated *A* (measured at 1000 µmol photons m^−2^ s^−1^ for both LL and HL leaves) to total leaf N concentration on a per-area basis. The leaf N fractions that were allocated into the major photosynthetic components (*P*_R_; *P*_B_, i.e., other Calvin cycle enzymes, ATP synthase, and electron carriers; and *P*_L_) were calculated according to [63], for which the specific leaf mass, N, and Chl concentrations (see below), and values of *J*_max_ and *V*_cmax_ (*C*_c_ basis) were used. The sum of *P*_R_ + *P*_B_ + *P*_L_ reflects the N fraction allocated into the photosynthetic machinery (*P*_P_), whereas the fraction of N partitioned into non-photosynthetic components (*P*_NP_) was quantified as *P*_NP_ = 1 − *P*_P_.

### 4.4. Metabolites

Leaf samples, collected at midday, were lyophilized at −48 °C and crushed in a ball mill, from which the concentrations of some metabolites were assessed. Briefly, pure methanol (700 μL) was added to a 10-mg sample of pulverized tissue, followed by incubation at 70 °C for 30 min. The mixture was then centrifuged (16,200× *g*, 5 min), and the supernatant was added to new tubes with the addition of chloroform (375 μL) and ultrapure water (600 μL). After new centrifugation (10,000× *g*, 10 min), the concentrations of hexoses (glucose and fructose) and sucrose were assessed in the soluble phase by a three-step reaction in which hexokinase, phosphoglucose isomerase and invertase were subsequently added to a reaction buffer containing ATP, NADH and glucose dehydrogenase (all from Sigma Aldrich). TAA was also quantified from the soluble phase using a standard curve with an equimolecular mixture of glycine, glutamic acid, phenylalanine, and arginine in 70% (*v*/*v*) aqueous ethanol. The methanol-insoluble pellet was used to quantify proteins and starch. The pellet was treated with KOH 0.2 M (1 mL), and then resuspended and incubated at 95 °C. Proteins were quantified afterward (Bradford method). Starch was determined after the sequential addition of acetic acid (2 M, 160 μL) and hexokinase in a buffer reaction as described above for sugars. Further details have been described elsewhere [64,65]. Chl (*a* + *b*) was extracted using aqueous acetone (80%, *v*/*v*) and quantified according to Wellburn [66].

### 4.5. Morphoanatomical Analyses

Samples from the middle portion of leaf blades (avoiding the midrib) were obtained, and then stored and processed for posterior analyses exactly as described elsewhere [67]. For each sample, five fields of view were used for measuring SD, SI, and SS. Magnifications were 10× for SD, and 20× for SI and SS. The SD was computed as the total number of stomata per area, and SI was calculated as the stomatal number-to-epidermal cell number ratio. The SS was calculated as guard cell length multiplied by the width of the guard cell pair according to Franks & Beerling [68]. The slides were photographed using a digital camera (Zeis AxioCan HRc, Göttingen, Germany) mounted on a light microscope (AX70 TRF, Olympus Optical, Tokyo, Japan); images were captured and analyzed using the Image-Pro Plus 4.5 software (Media Cybernetics, Silver Spring, ML, USA). Additionally, *g*_wmax_ was calculated based on anatomical traits, exactly as described in [12].

### 4.6. Growth Attributes

At the end of the experiment, growth traits, i.e., plant height, leaf area, and diameter of the orthotropic branch (5 cm above the ground) were quantified. TLA was measured using a leaf area integrator (Model 3100, LI-Cor, Lincoln, NE, USA). Roots were carefully washed with tap water over a 0.5 mm sieve. Additionally, leaves (including those that had eventually abscised during the experiment), stems (orthotropic and plagiotropic branches), and roots were oven-dried at 70 °C for 72 h to obtain their respective dry masses. Based on these data, TB, LMR, SMR (which was fractionated into OMR and PMR), and LAR were calculated [48].

### 4.7. Statistical Analysis

Data were subjected to a two-way ANOVA to assess the effects of factors *C*_a_ and light, and their interaction, using the following model:y_ijk_ = μ + α_i_ + γ_j_ + (αγ)_ij_ + e_ijk_
where y_ijk_ is the observation value of *ijk*^th^ experimental unit; μ is the overall mean; α_i_ is the fixed effect of *i*th level of factor *C*_a_; γ_j_ is the fixed effect of *j*th level of factor light; (αγ)_ij_ is the fixed effect of interaction between *i*th level of factor *C*_a_ and *j*th level of factor light; e_ijk_ is the random error of *ijk*th experimental unit, with e_ijk_~*N*(0, $\sigma^{2}$). Afterward, a *t* test (*p* ≤ 0.05) was performed for mean comparisons using the SISVAR software version 5.6 [69]. Data were expressed as means ± standard error.

## 5. Conclusions

We demonstrate that an e*C*_a_ improves the growth and photosynthetic performance of coffee plants, but these improvements decrease at deep shade. Coffee plants are apparently highly suited for a changing climate characterized by a continuing elevation of *C*_a_, especially if the light is non-limiting. Some findings that support this idea are (i) greater *g*_s_ at e*C*_a_ in parallel with decreased diffusive limitations to photosynthesis, (ii) greater *g*_s_ during sharp HL-to-LL transitions, (iii) greater leaf N pools and PNUE, (iv) lack of photosynthetic acclimation, even when sink demand is severely restricted, (v) greater biomass partitioning to roots, which can improve water and nutrient acquisition, and (vi) earlier branching, which may be reflected in improved plant fitness and crop yield. In summary, our novel and timely findings provide relevant information on the roles of e*C*_a_ to improve coffee growth and yield in a changing climate scenario.

## Figures and Tables

**Figure 1 plants-12-01479-f001:**
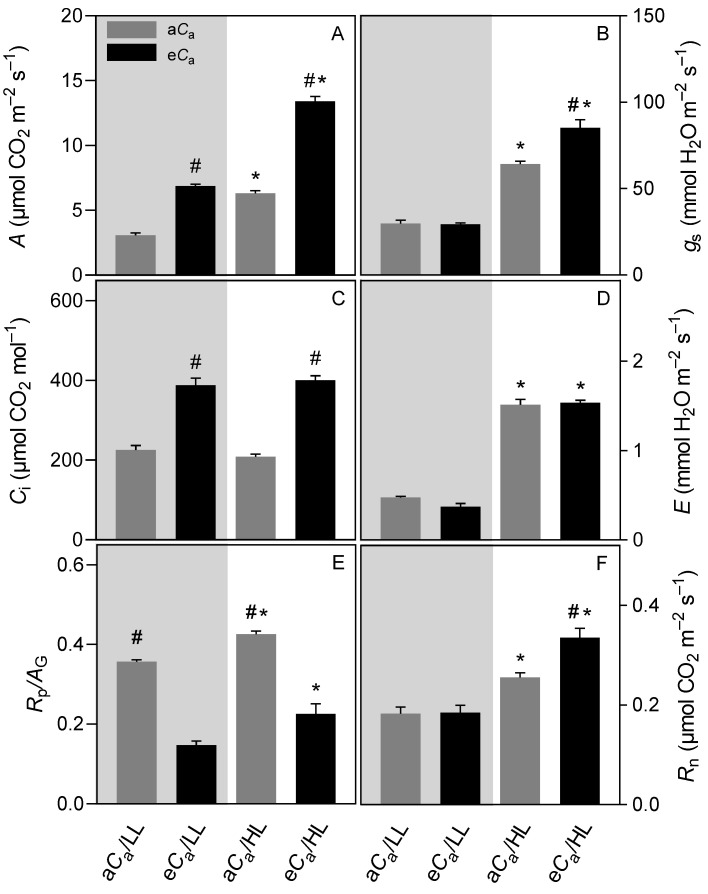
The effect of light intensity [low (grey area) or high (white area) light] and CO_2_ supply [ambient (a*C*_a_) or elevated (e*C*_a_) supply] on (**A**) net carbon assimilation rate (*A*), (**B**) stomatal conductance (*g*_s_), (**C**) internal CO_2_ concentration (*C*_i_), (**D**) transpiration rate (*E*), (**E**) photorespiration-to-gross photosynthetic rate ratio (*R*_p_/*A*_G_), and (**F**) nocturnal respiration (*R*_n_). Asterisk (*), when shown, indicates differences between light regimes within the same CO_2_ supply. Pound (#), when shown, indicates differences between CO_2_ treatments within the same light regime (*p* ≤ 0.05, *F* test, *n* = 6 ± SE).

**Figure 2 plants-12-01479-f002:**
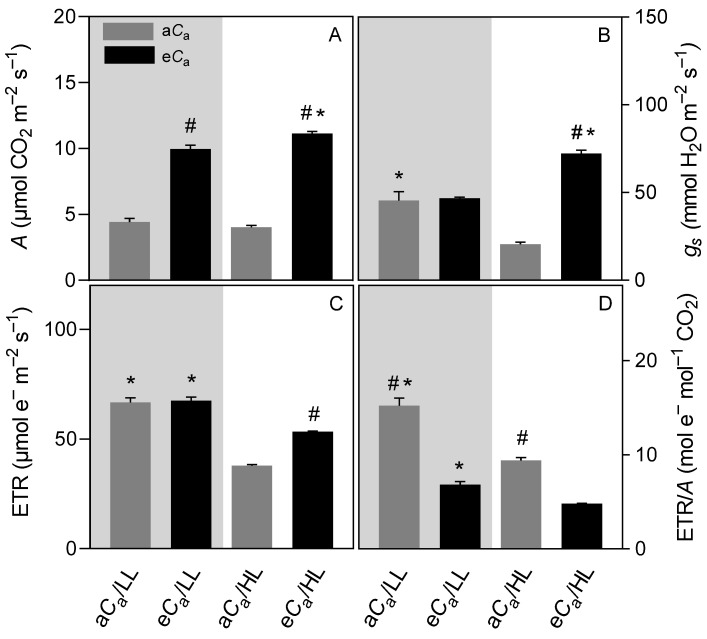
The effect of light transitions (grey area: plants grown at low light but evaluated at high light, 1000 µmol photons m^−2^ s^−1^; white area: plants grown at high light but evaluated at low light, 200 µmol photons m^−2^ s^−1^) and CO_2_ supply [ambient (a*C*_a_) or elevated (e*C*_a_) supply] on the (**A**) net carbon assimilation rate (*A*), (**B**) stomatal conductance (*g*_s_), (**C**) electron transport rate (ETR), and (**D**) ETR/*A* ratio. Asterisk (*), when shown, indicates differences between light regimes within the same CO_2_ supply. Pound (#), when shown, indicates differences between CO_2_ treatments within the same light regime (*p* ≤ 0.05, *F* test, *n* = 6 ± SE).

**Figure 3 plants-12-01479-f003:**
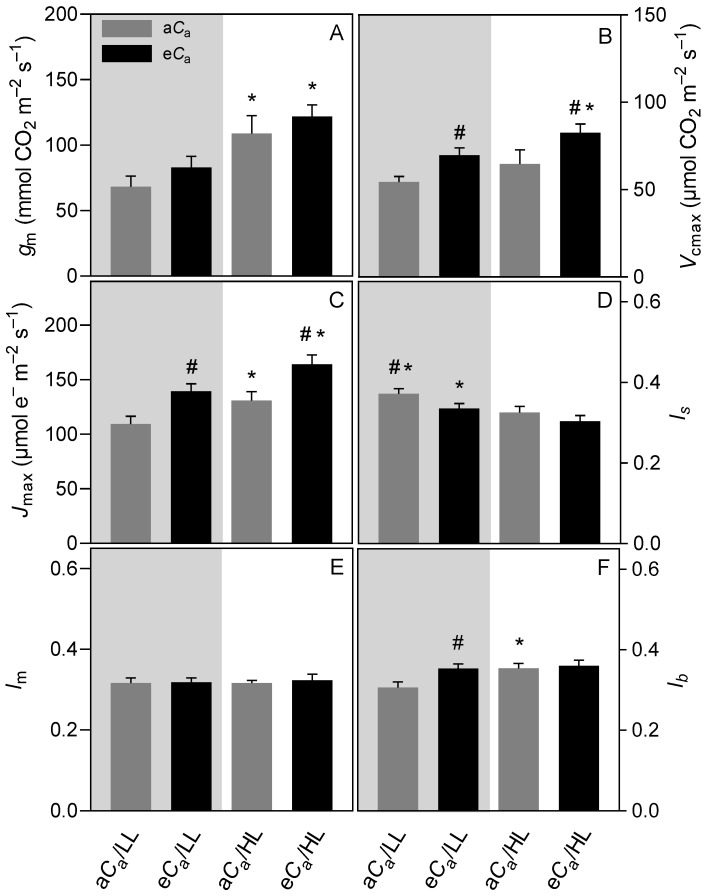
The effect of light intensity [low (grey area) or high (white area) light] and CO_2_ supply [ambient (a*C*_a_) or elevated (e*C*_a_) supply] on photosynthetic traits of coffee plants: (**A**) mesophyll conductance to CO_2_ (*g*_m_), (**B**) maximum apparent carboxylation capacity (*V*_cmax_), and (**C**) maximum rate of carboxylation limited by electron transport (*J*_max_), both on a chloroplast [CO_2_] basis, and the functional components of the overall limitations to photosynthesis [(**D**) stomatal (*l*_s_), (**E**) mesophyll (*l*_m_), and (**F**) biochemical (*l*_b_) limitations]. Asterisk (*), when shown, indicates differences between light regimes within the same CO_2_ supply. Pound (#), when shown, indicates differences between CO_2_ treatments within the same light regime (*p* ≤ 0.05, *F* test, *n* = 6 ± SE).

**Table 1 plants-12-01479-t001:** The effect of light intensity [low (LL) or high (HL) light] and CO_2_ supply [ambient (a*C*_a_) or elevated (e*C*_a_)] on nitrogen traits of coffee plants: foliar N on a per mass (N_m_) and area (N_a_) basis, the leaf N fractions that were allocated into the major photosynthetic components [(RuBisCO (*P*_R_), bioenergetics (*P*_B_), and thylakoid light-harvesting components (*P*_L_)], the N fraction that was allocated into the photosynthetic machinery (*P*_P_), and the fraction of N that was partitioned into non-photosynthetic components (*P*_NP_). The photosynthetic N-use efficiency is also shown. Asterisk (*), when shown, indicates differences between light regimes within the same CO_2_ supply. Pound (#), when shown, indicates differences between CO_2_ treatments within the same light regime (*p* ≤ 0.05, *F* test, *n* = 6 ± SE).

Parameters	LL		HL	
	a*C*_a_	e*C*_a_	a*C*_a_	e*C*_a_
N_m_ (g kg^−1^)	30.6 ± 0.01	32.8 ± 0.1	32.6 ± 0.1	37.9 ± 0.1 #*
N_a_ (g m^−2^)	1.56 ± 0.04	1.65 ± 0.08	1.73 ± 0.08	2. 16 ± 0.08 #*
*P* _R_	0.26 ± 0.00	0.29 ± 0.02 #	0.22 ± 0.01 *	0.23 ± 0.01 *
*P* _B_	0.036 ± 0.003	0.037 ± 0.002	0.026 ± 0.001 *	0.032 ± 0.001 #*
*P* _L_	0.18 ± 0.00	0.19 ± 0.01	0.16 ± 0.01	0.15 ± 0.01 *
*P* _P_	0.47 ± 0.01	0.52 ± 0.02 #	0.41 ± 0.02 *	0.42 ± 0.02 *
*P* _NP_	0.52 ± 0.01	0.48 ± 0.01 #	0.59 ± 0.02 *	0.58 ± 0.02 *
PNUE (µmol CO_2_ g^−1^ N)	2.85 ± 0.12	6.08 ± 0.27 #	3.81 ± 0.15 *	6.41 ± 0.34 #

**Table 2 plants-12-01479-t002:** The effect of light intensity [low (LL) or high (HL) light] and CO_2_ supply [ambient (a*C*_a_) or elevated (e*C*_a_)] on foliar pigments and metabolites (on a dry weight basis) of coffee plants: total chlorophylls (Chl *a* + *b*), starch, glucose, fructose, sucrose, nonstructural carbohydrates (NSC), total amino acids (TAA), and proteins. The Chl/N ratio is also presented. When shown, asterisk (*) indicates differences between light regimes within the same CO_2_ supply, whereas hash (#) indicates differences between CO_2_ treatments within the same light regime (*p* ≤ 0.05, *t* test, *n* = 6 ± SE).

Parameters	LL		HL	
	a*C*_a_	e*C*_a_	a*C*_a_	e*C*_a_
Chl *a + b* (mg g^−1^)	7.8 ± 0.3	6.9 ± 0.3	6.4 ± 0.9 *	4.5 ± 0.2 #*
Chl/N (mmol mol^−1^)	2.5 ± 0.1	2.1 ± 0.1 #	2.0 ± 0.2	1.2 ± 0.1 #*
Strach (µmol eq. glucose g^−1^)	568 ± 9	571 ± 10	684 ± 28 *	641 ± 10 #*
Glucose (µmol g^−1^)	24.9 ± 1.2	23.9 ± 0.6	28.8 ± 3.5 *	34.7 ± 1.5 #*
Fructose (µmol g^−1^)	29.6 ± 0.6	26.3 ± 0.6 #	33.5 ± 1.6 *	35.9 ± 0.5 #*
Sucrose (µmol g^−1^)	64.9 ± 1.9	75.3 ± 2.1 #	63.3 ± 1.9	79.5 ± 1.8 #
NSC	687 ± 11	696 ± 12	810 ± 11 *	791 ± 9 *
TAA (µmol g^−1^)	11.3 ± 1.5	20.3 ± 0.9 #	18.7 ± 3.29 *	29.6 ± 1.2 #*
Proteins (mg g^−1^)	95 ± 5	114 ± 4 #	100 ± 12	121 ± 4 #

**Table 3 plants-12-01479-t003:** The effect of light intensity [low (LL) or high (HL) light] and CO_2_ supply [ambient (a*C*_a_) or elevated (e*C*_a_)] on stomatal density (SD), stomatal index (SI), stomatal size (SS), and maximum (theoretical) stomatal conductance to water vapor (*g*_wmax_) of coffee plants. When shown, asterisk (*) indicates differences between light regimes within the same CO_2_ supply, whereas hash (#) indicates differences between CO_2_ treatments within the same light regime (*p* ≤ 0.05, *t* test, *n* = 6 ± SE).

Parameters	LL		HL	
	a*C*_a_	e*C*_a_	a*C*_a_	e*C*_a_
SD (mm^−2^)	83 ± 6	87 ± 5	167 ± 10 *	186 ± 6 *
SI (%)	15.5 ± 0.4	16.0 ± 0.3	18.8 ± 0.5 *	20.1 ± 0.7 *
SS (µm^2^)	172 ± 11	164 ± 12	161 ± 13	193 ± 9
*g*_wmax_ (mol m^−2^ s^−1^)	0.68 ± 0.05	0.70 ± 0.05	1.32 ± 0.08 *	1.61 ± 0.02 #*

**Table 4 plants-12-01479-t004:** The effect of light intensity [low (LL) or high (HL)] and CO_2_ supply [ambient (a*C*_a_) or elevated (e*C*_a_)] on growth traits of coffee plants: height, total biomass (TB), height, total leaf area (TLA), leaf mass ratio (LMR), stem mass ratio (SMR), orthotropic branch mass ratio (OMR), plagiotropic branch mass ratio (PMR), root mass ratio (RMR), and leaf area ratio (LAR). When shown, asterisk (*) indicates differences between light regimes within the same CO_2_ supply, whereas hash (#) indicates differences between CO_2_ treatments within the same light regime (*p* ≤ 0.05, *t* test, *n* = 6 ± SE).

Parameters	LL		HL	
	a*C*_a_	e*C*_a_	a*C*_a_	e*C*_a_
TB (g)	25.3 ± 1.1	38.7 ± 1.3 #	119 ± 7.7 *	240 ± 7.4 #*
Height (cm)	44 ± 2.1	55 ± 1.0 #	84 ± 3.6 *	103 ± 0.7 #*
TLA (m^2^)	0.26 ± 0.01	0.37 ± 0.01 #	0.78 ± 0.05 *	1.26 ± 0.04 #*
LMR	0.61 ± 0.02	0.55 ± 0.01 #	0.65 ± 0.02	0.52 ± 0.02 #
SMR	0.27 ± 0.02	0.27 ± 0.01	0.24 ± 0.01	0.26 ± 0.01
OMR	0.21 ± 0.01	0.21 ± 0.01	0.17 ± 0.01 *	0.18 ± 0.01
PMR	0.056 ± 0.004	0.068 ± 0.005 #	0.070 ± 0.039 *	0.082 ± 0.003 #*
RMR	0.12 ± 0.01	0.18 ± 0.02 #	0.11 ± 0.01	0.22 ± 0.02 #
LAR (m^2^ kg^−1^)	10.5 ± 0.8	9.5 ± 0.4	6.4 ± 0.2 *	5.3 ± 0.3 *

## Data Availability

Not applicable.

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
