# Peer review of "Growth and Leaf Gas Exchange Upregulation by Elevated [CO2] Is Light Dependent in Coffee Plants"

_plants, 2023, doi:10.3390/plants12071479_

Round 1
Reviewer 1 Report (Previous Reviewer 2)
In all figures, the X axis needs to be labelled.
Author Response
We suppose that the reviewer#1 have appreciated our MS. His/her only comments were “In all figures, the X axis needs to be labelled”. However, graphs were presented as histograms and, from our perspective, no need to label the x-axis is necessary. Please note that legends for the proper understanding of the data outlined in each graph has been properly provided. In this context, we did not proceed with the reviewer’s suggestion.
Reviewer 2 Report (New Reviewer)
The paper titled: “Growth and leaf gas exchange upregulation by elevated [CO2] is light dependent in coffee plants” submitted by the authors de Souza et al., studied how light levels and Ca could affect the stomatal function, the relative contributions of photochemical, diffusional, and biochemical limitations of photosynthesis at the leaf level, the N partitioning into the major components of the photosynthetic apparatus, as well as the coffee growth performance.
There are some things need to be addressed before the publishing of this paper:
1. In the introduction:
- The novelty of the work had been clearly highlighted.
- The introduction is very long and should be reduced to four main paragraphs showing the necessity of this work.
- Try to merge the first the paragraph into other paragraphs and reduce the size of each paragraph to the necessary data.
2. The results are well presented.
3. The discussion: There are old references need to be removed. Also in the introduction part such as :
Woodward, F.I. Stomatal numbers are sensitive to increases in CO2 from pre-industrial levels. Nature, 1987, 327, 617–618.
All old references than 2010 should be removed and focus on the last 5 years.
4. The materials and methods part
- Lines 448-449, the source of the seedlings or seeds should be declared as a voucher deposit and the person whom identified the cultivar. Also if there is a figure for the cultivar, it will much better.
- Basic soil analyses should be declared for the soil used for the growing of the plants
- In all abbreviations used please spell all these abbreviations within text at first mention to smooth the reading of this paper. I see that you used a list of abbreviations up there, but this still necessity.
- Details of the methodology of the quantification of carbohydrates, i.e., glucose, fructose, sucrose, and starch (using enzymatic assays), TAA, and proteins should be declared here.
5. The conclusion need to show the major results found in the study and the prospect of the work.
6. References: Old references (such as those in 2001) need to be removed, please try to focus on the last 5 years.
I give you major revision.
Author Response
. In the introduction:
- The novelty of the work had been clearly highlighted.
Response: We are very happy with this nice comment.
- The introduction is very long and should be reduced to four main paragraphs showing the necessity of this work.
- Try to merge the first the paragraph into other paragraphs and reduce the size of each paragraph to the necessary data.
Response: We have shortened the Introduction and have merged the 1st paragraph with the 2nd one. The current version has now 4 paragraphs.
- The results are well presented.
Response: We are very happy with this nice comment.
- The discussion: There are old references need to be removed. Also in the introduction part such as :
Woodward, F.I. Stomatal numbers are sensitive to increases in CO2 from pre-industrial levels. Nature, 1987, 327, 617–618.
All old references than 2010 should be removed and focus on the last 5 years.
Response: We clearly understand this criticism however we have a bit different point of view regarding the use of references that were published before 2010. Some classical, seminal (e.g. meta analyses articles on the plant responses to elevated CO2) papers should/must be cited, from our perspective. Scientific credit must be acknowledged to pioneer references. In any case, to satisfy the reviewer’s uncertainty we have deleted all the references that were published before 2000. Considering a total of 50 references that were quoted in Introduction and Discussion, only 9 of them were published before 2012 (but after 2000), and 27 (more than half) were published in the last 5 years.
- The materials and methods part
- Lines 448-449, the source of the seedlings or seeds should be declared as a voucher deposit and the person whom identified the cultivar. Also if there is a figure for the cultivar, it will much better.
Response: We did not clearly understand this suggestion. We believe that the suggestion, as long as we have understood it, is out of the scope of our paper and it does not add any important information to understand the current study. In any case, we have added information regarding as how we have obtained the seedlings. We should also emphasize that the cv. Catuaí Vermelho IAC 44 was developed by the “Instituto Agronômico de Campinas” as abbreviated by IAC. This cultivar was commercially launched in 1972 and registered in the Brazilian Register of Cultivars in 1999. It is one of the main coffee cultivars grown in Brazil.
Basic soil analyses should be declared for the soil used for the growing of the plants
Response: details on plant soil analysis have been provided as requested.
- In all abbreviations used please spell all these abbreviations within text at first mention to smooth the reading of this paper. I see that you used a list of abbreviations up there, but this still necessity.
Response: This is an important criticism. However, we had already spelled out all the abbreviations throughout the text following their first mention.
- Details of the methodology of the quantification of carbohydrates, i.e., glucose, fructose, sucrose, and starch (using enzymatic assays), TAA, and proteins should be declared here.
Response: We have now expanded this section as requested. Hopefully we have properly addressed the reviewer uncertainty.
- The conclusion need to show the major results found in the study and the prospect of the work.
Response: We clearly understand this criticism. Nonetheless, we believe that “Conclusions” have been written exactly following this useful suggestion. We therefore did not modify this section.
- References: Old references (such as those in 2001) need to be removed, please try to focus on the last 5 years.
Response: Please see our above comments.
Round 2
Reviewer 2 Report (New Reviewer)
Accepted for me
Author Response
Your recommendation to label the x-axis of figures has now been made accordingly.
This manuscript is a resubmission of an earlier submission. The following is a list of the peer review reports and author responses from that submission.
Round 1
Reviewer 1 Report
The authors aimed to study the acclimation of coffee to slight elevations in the atmospheric CO2 concentration with parallel changes in the surrounding light intensity. Some phenotypical, physiological and metabolite parameters were measured and the authors tried to make some correlations out of these data in order to support their hypothesis. However, and to be honest, the study does not go much beyond what is already know in the field and is rather descriptive in its current state. The only partially new finding might be that coffee seems to have a bit higher stomatal conductance at slightly elevated CO2 compared to other species, which is very likely the key explanation for some increases in photosynthesis and growth parameters under the studied conditions. Despite the study itself, in terms of the experiments carried out and the statistical evaluation of the data, seems to be conducted fairly well, I would have welcomed some affords explaining why coffee might have some reduced stomatal closure at elevated CO2 concentrations. It would also be a clear benefit to determine the range of CO2 concentrations that coffee can accept without displaying major decreases in gs. Finally, apart from the fact that there are not that much groups working on coffee, it seems that the authors tried their best to cite as much of their own published papers as possible, especially the senior author, which has to be more representative for the field of research in general.
Reviewer 2 Report
Manuscript Review
Growth and leaf gas exchange upregulation by elevated [CO2] is light dependent in coffee plants
By Antonio F. De Souza, Uéliton S. De Oliveira, Leonardo A. Oliveira, Pablo H.N. De Carvalho, Moab T. De Andrade, Talitha S. Pereira, Carlos C. Gomes Júnior, Amanda A. Cardoso, José D.C. Ramalho, Samuel C.V. Martins, and Fábio M. DaMatta
Submitted to Plants (Plants-1996414)
In this paper, the authors examined the interactive effects of carbon dioxide and light intensity on growth and photosynthetic performance of coffee plants. They have shown that elevated carbon dioxide improves coffee growth and photosynthesis. This manuscript requires proper revisions before it can be considered for publication.
Abstract
Line 28. It is better to consider the effects of elevated carbon dioxide on atmospheric temperature and discuss coffee performance under both elevated carbon dioxide and high temperature. This point should be mentioned in the abstract briefly and discussed in the Discussion in details.
Introduction
Line 93. Is it agroforest systems or agroforestry systems?
Line 133. A clear hypothesis as well as the study objectives should be provided here.
Results
Table 1. It is better to provide F values, which are followed by P values, shown as asterisks.
Figure 1. The use of asterisk and pound is confusing. It is better to use lowercase and uppercase letters. It should be done for all figures, and letters should be used above all bars.
Figure 1. In the legend, aCa should be written before eCa.
Figures 2 and 3. Please see the suggestion for Figure 1.
Table 2. It is better to use lowercase (a,b,...) and uppercase (A, B, ...) letters to show differences between treatments. All values should be followed by letters.
Tables 3, 4, and 5. Please see the suggestion for Table 2.
Discussion
It is better to discuss the possible effects of high temperature and drought stress on plant traits.
Line 511. Is it "Nitrogen use efficiency"?
Appendix A. Supplementary data
Suppl. Table 1. Please see the suggestion for Table 2.
References
Lines 657-659. The format of this reference needs to be corrected.
Reviewer 3 Report
This work is aimed at understanding the impact of elevated atmospheric CO2 on the growth and productivity of coffee plants. The researchers grew young coffee plants from rooted cuttings in open top chambers to control CO2 at two levels of approximately 437 and 705 µmol mol-1. The open-top chambers were situated inside of a greenhouse. The high light treatment plants received the ambient light levels in the greenhouse, although presumably somewhat attenuated by the chamber, while the low-light treatment plants were shaded approximately 90% with neutral density shade cloth. The researchers made extensive gas exchange and chlorophyll fluorescence measurements as well as plant growth, nitrogen, and nonstructural carbohydrate measurements. Using these results, the authors draw some conclusions regarding the impact of ambient PAR and CO2 on how coffee plants will respond to future atmospheric conditions.
Notes to the authors
Table 1 covers several pages of the published manuscript, and is focussed solely on the statistical analysis results. This is distracting to the reader. Given that the significant results are stated elsewhere this table could be placed in the supplementary material.
Light treatments, measured as daily integrated PFD reported to be 1 and 20 mol m-2 day-1. Even the high light treatment was low compared with outdoor field-grown plants (expected to be 40-60 mol m-2 day-1). The authors should acknowledge that this "high-light" treatment is more moderate relative to agricultural conditions.
Section headings are full sentences describing the authors conclusions rather than a label of the section topic. e.g.,
- 3.1 Stomatal conductance
- 3.2 Photosynthetic performance
- 3.3 N use efficiency and partitioning
- 3.4 Growth and Biomass partitioning
I will note that many readers will see that this design suffers from pseudoreplication. Each individual chamber could be considered a replicate, not each potted plant. I note this but do not reject the science on this basis because I understand that researchers have constraints on time, equipment, and budgets. In future studies, however, the researchers may want to consider this in their experimental design.